# An Attenuated HSV-1-Derived Malaria Vaccine Expressing Liver-Stage Exported Proteins Induces Sterilizing Protection against Infectious Sporozoite Challenge

**DOI:** 10.3390/vaccines10020300

**Published:** 2022-02-16

**Authors:** Paul J. F. Rider, Mohd Kamil, Ilknur Yilmaz, Habibe N. Atmaca, Merve Kalkan-Yazici, Mehmet Ziya Doymaz, Konstantin G. Kousoulas, Ahmed S. I. Aly

**Affiliations:** 1Division of Biotechnology and Molecular Medicine, Louisiana State University, Baton Rouge, LA 70803, USA; prider@lsu.edu (P.J.F.R.); vtgusk@lsu.edu (K.G.K.); 2Department of Pathobiological Sciences, Louisiana State University, Baton Rouge, LA 70803, USA; 3Aly Lab, Beykoz Institute of Life Sciences and Biotechnology, Bezmialem Vakif University, Istanbul 34820, Turkey; mohdkamil54@gmail.com (M.K.); ilknur.ylmzgen@gmail.com (I.Y.); atmacahabibe@gmail.com (H.N.A.); 4Microbiology Lab, Beykoz Institute of Life Sciences and Biotechnology, Bezmialem Vakif University, Istanbul 34820, Turkey; mkyazici@bezmialem.edu.tr (M.K.-Y.); mzdoymaz@bezmialem.edu.tr (M.Z.D.)

**Keywords:** malaria vaccine, VC2, liver stage, EXP1, UIS3, TMP21, intramuscular, subcutaneous, HSV-1, viral vector

## Abstract

Here, we present the construction of an attenuated herpes simplex virus type-1 (HSV-1)-vectored vaccine, expressing three liver-stage (LS) malaria parasite exported proteins (EXP1, UIS3 and TMP21) as fusion proteins with the VP26 viral capsid protein. Intramuscular and subcutaneous immunizations of mice with a pooled vaccine, composed of the three attenuated virus strains expressing each LS antigen, induced sterile protection against the intravenous challenge of *Plasmodium yoelii* 17X-NL salivary gland sporozoites. Our data suggest that this malaria vaccine may be effective in preventing malaria parasite infection using practical routes of immunization in humans.

**Lay summary:** We developed a live attenuated HSV-1-vectored vaccine, expressing the EXP1, UIS3, and TMP21 malaria parasite liver-stage exported antigens as fusion proteins with the VP26 viral tegument protein. Intramuscular and subcutaneous immunizations resulted in 100% sterile protection in mice challenged with *P. yoelii* 17X-NL infectious sporozoites.

## 1. Background

While recent intervention strategies have led to a decline in the number of malaria infections and deaths worldwide, malaria continues to kill more than 400,000 individuals yearly. Intervention strategies and anti-malaria drugs have had a significant impact on the number of malaria infections and associated illnesses. However, recently, the World Health Organization (WHO) has reported that the rate of decline in malaria cases has slowed to 2% from 2015 to 2019, whereas there was a 27% decline between 2000 and 2015 [1]. It is clear that there is an urgent need for a safe, effective malaria vaccine.

Human malaria is caused by five different species of *Plasmodium* parasites, of which *P. falciparum* has the highest mortality rate. In mammals, the life cycle of the parasite is initiated when sporozoites are introduced into the bloodstream after the host is bitten by a malaria-infected mosquito. The sporozoites then travel to the liver, where they grow into a liver stage (LS) and mature into merozoites that are released into the bloodstream to invade erythrocytes and cause clinical manifestations of malaria infection. Malaria is clinically silent prior to the emergence of merozoites from the liver. As such, the pre-erythrocytic silent stage is a primary target for vaccine development. Despite decades of efforts towards developing an effective vaccine, the pre-erythrocytic Circumsporozoite protein (CSP) based the RTS,S subunit vaccine is the most advanced vaccine candidate [2]. Unfortunately, recent phase 3 clinical trials of RTS,S showed modest to low efficacy. Furthermore, the efficacy of the vaccine waned over time [3].

In this study, we used the novel HSV-1-derived viral vaccine vector VC2 to express three different malaria parasite LS antigens. VC2 is safe and immunogenic in mice, guinea pigs, and non-human primates [4]. To produce a malaria vaccine, we selected the highly conserved malaria parasite proteins EXP1 (exported protein 1), TMP21 (transmembrane protein 21) and UIS3 (upregulated in infectious sporozoites), as they were the best LS exported protein candidates that showed, individually, the best results in protecting immunized mice against an infectious sporozoite challenge, and eliciting CD8 T cells in a previous DNA immunization study from our lab [5]. However, the best protection against a challenge was achieved when the three proteins were combined in intramuscular DNA immunization [5]; therefore, here, they were selected in a combination of three recombinant viral vaccines, each expressing one of the selected LS exported antigens. We specifically chose highly conserved and low-molecular-mass malaria parasite LS exported antigens with the rationale that (i) the export of these antigens at or beyond the parasitophorous vacuolar membrane (PVM) makes them more likely to be presented via the MHC I of infected hepatocytes; (ii) these proteins are of relatively low molecular mass, enabling their efficient expression as fusion proteins with the HSV-1 VP26 tegument protein; (iii) these proteins are expressed in *Plasmodium* species during early to mid LS development, allowing for the possibility that effector CD8+ T cells could target infected hepatocytes before merozoites are formed, preventing clinical malaria [5,6]. To this end, we generated three unique HSV-1 viruses expressing each of the malaria antigens. These vaccines were pooled and used to vaccinate mice prior to intravenous challenge with salivary gland sporozoites. Herein, we show that this approach generates sterile protection against a virulent *Plasmodium* sporozoite challenge in a murine model. These data identify EXP1, TMP21, and UIS3 as suitable targets for future malaria vaccine efforts and support the further development of the HSV-1 (VC2)-vectored malaria vaccine approach.

## 2. Methods

### 2.1. Animals and Parasites

Six-to-eight-week-old female BALB/c mice were purchased from the animal service facility of Bezmialem Vakif University. *P. yoelii* 17X-NL strain was stored as frozen stocks at −150 °C. Freshly thawed parasites were injected into donor-naive mice for mosquito feeding to obtain sporozoites. *Anopheles stephensi* mosquitos were infected by allowing them to feed on parasite-infected female BALB/c mice as described previously [7].

### 2.2. Viral Vector Construction

For the generation of VC2-derived malaria vaccines, we chose the liver-stage antigens EXP1 (XP_724715.1), TMP21 (XP_727064.1) and UIS3 (XP_022812792.1). Red-mediated recombination of VC2 bacterial artificial chromosome (BAC) [8] was used to independently generate the following three vaccines: VC2-EXP1, VC2-TMP21, and VC2-UIS3. Using this technique we fused each malaria protein sequence to VP26 in a manner previously published by Desai et al., for fusion of GFP to VP26 [9]. Transmembrane domains of malaria proteins were predicted and removed from the final sequences used to generate synthetic sequences for insertion. All recombinant BACs were sequenced to confirm the correct sequence of fusion proteins.

### 2.3. Cell Culture and Viral Propagation

Vero cells were obtained from ATCC and maintained in DMEM medium (Gibco, cat#11965092) supplemented with 10% fetal bovine serum (Gibco, cat#10082147) and 100 U of penicillin–streptomycin/mL (Gibco, cat#15070063). To prepare purified viruses, Vero cells were cultured in T-75 cell culture flask. After 24 h, at about 90% confluency, cells were infected with the virus using 0.01 MOI (multiplicity of infection) and incubated at 37 °C in 5% CO_2_. At maximum cytopathic effect the infected cells were harvested by scraping, and supernatants were collected and cleared by low-speed centrifugation at 1500 rpm for 10 min at room temperature. The viruses collected from culture supernatant were aliquoted and stored at −80 °C. Virus titer was determined by calculating plaque forming units in Vero cells.

### 2.4. Immunoblotting

Vero cells were infected with VC2, VC2-EXP1, VC2-TMP21 and VC2-UIS3 at an MOI of 0.01. Forty-eight hours post infection, cell lysates were collected in Laemmli buffer with protease inhibitor cocktail and subjected to immunoblotting. An amount of 20 µL of lysate was electrophoresed on 12% sodium dodecyl sulfate polyacrylamide (SDS-PAGE) gel and transferred to PVDF membrane. The membrane was probed with anti-VP26 antibody (a kind gift from Prashant Desai).

### 2.5. Vaccination

Six-week-old female BALB/c mice were used. A titer of 1 × 10^6^ plaque forming units (PFU) of each vaccine candidate was used to immunize mice. Vaccines VC2-EXP1, VC2-UIS3 and VC2-TMP21 were pooled and administered to mice in a total volume of 150 µL with RPMI media (Gibco). Intramuscular (IM) (*n* = 8 mice per group) and subcutaneous (SC) (*n* = 9 mice per group) routes were used to immunize mice. Three immunizations were performed at three-week intervals between prime (0 day), first boost (21 days) and second boost (42 days). Only VC2 in 150 µL at a titer of 1 × 10^6^ PFU/mL (*n* = 5 mice per group) was used as a control.

### 2.6. Parasite Challenge

Mice were intravenously challenged with *P. yoelii* 17X-NL salivary gland sporozoites resuspended in incomplete RPMI media. Sporozoites were prepared as described earlier [7]. Twelve weeks after the final immunization, each mouse was injected via the tail vein with 150 µL of incomplete RPMI media, containing 500 salivary gland sporozoites. A group of seven naive mice was also used as a control in parasite challenge study. After parasite challenge, infection was monitored from day 3 to day 15 post challenge by Giemsa staining of tail vein thin blood smears. Protection was defined as the complete absence of erythrocytic-stage parasitemia on day 15 post challenge.

## 3. Results

### 3.1. Construction of Recombinant VC2 Virus Expressing Malaria Antigens

We chose three malaria LS antigens (EXP1, TMP21, and UIS3) to be expressed as fusion proteins with the HSV-1 minor capsid protein VP26 (Figure 1A). We generated the following three recombinant viruses, each with a different LS antigen fused to VP26: VC2-EXP1, VC2-UIS3, and VC2-TMP21. Western blotting, using an anti-VP26 antibody, demonstrated the predicted fusion to respective malaria LS proteins (Figure 1B). The growth assay showed no significant differences in the viral titers of the vaccines compared to the parental VC2 virus (Figure 1C).

### 3.2. VC2-EXP1, VC2-TMP21, and VC2-UIS3 Pooled Vaccine Demonstrates Sterile Protection in Immunized Mice

A challenge with *Plasmodium* sporozoite is the best and most effective way to analyze the effectiveness of any pre-erythrocytic-stage malaria vaccine. In our immunization experiments, we used intramuscular (*n* = 8 mice per group) and subcutaneous (*n* = 9 mice per group) routes of immunization to immunize mice. A total of three immunizations were performed, each at three-week intervals. Twelve weeks after the final immunization, mice were intravenously challenged with 500 *P. yoelii* 17X-NL salivary gland sporozoites. A group of naive mice (*n* = 5 mice), without any prior injections, was also included in the challenge study. Complete sterile protection was observed in 17 out of 17 mice (both immunized groups) until day 15 post sporozoite challenge (Table 1). However, all the naive control mice or mice immunized with VC2 showed blood-stage infection by day 4 post sporozoite challenge.

## 4. Discussion

Herein, we report the development of a highly efficacious trivalent malaria vaccine using the attenuated HSV-1 (VC2) vaccine vector. Research groups working on malaria vaccines are mainly targeting the pre-erythrocytic, [10,11] erythrocytic [12] and transmission blocking [13] stages, but none of them have presented an effective vaccine that can induce complete sterile protection against malaria parasite infection or transmission. The most successful malaria vaccine thus far is the pre-erythrocytic-stage subunit malaria vaccine RTS,S [2]. Unfortunately, the RTS,S has not realized its promise, as its efficacy is very low and protective immune responses have waned over time. Decades of efforts have concentrated heavily on neutralizing the agile sporozoites in the blood stream within this very limited window of time before it reaches the liver. Moreover, sporozoites represent invasive-stage parasites that use their surface proteins as tools to traverse and invade host cells, but do not express those surface antigens once they invade a replication-permissive hepatocyte. Furthermore, we reasoned that a vaccine possessing a combination of antigens that are expressed in the liver stages would generate a more potent protective immune response against the intra-hepatocytic liver-stage parasites.

We selected the HSV-1 viral minor capsid protein VP26 to fuse with malaria antigens, as it can be fused to a variety of proteins without affecting virion assembly and replication. Specifically, the HSV-1 VP26 tegument protein has been fused with different fluorescent proteins for virus tracking experiments, without exhibiting any defects [9]. VP26 is present at nearly 1000 copies per virion [14], estimating that an inoculum of 1 × 10^6^ PFU will deliver 10^9^ antigens in a single dose of vaccine. The actual VP26 protein load may be much higher, since the PFU projection does not take into account non-infectious particles, which are expected to be present in a 10:1–100:1 particle-to-PFU ratio, which may increase the VP26 numbers to as high as 10^11^ molecules. Furthermore, as a replicating virus, the amount of VP26-fused antigens would be expected to increase with viral replication.

Our data clearly show that our VC2-vectored vaccine could be effectively used by both well-accepted IM and/or SC routes. While the use of viral vectors for vaccination against infectious diseases is not novel, we believe that our HSV-1 (VC2) vector is unique. The safety of the virus has been demonstrated in ocular infections and in infections of SCID mice [15]. The expression of malaria antigens as fusion proteins with the VP26 viral protein allows the antigens to be included in the virion particle, as well as to be expressed during lytic infection. In this regard, it is likely that these viral antigens are available for both MHCI- and MHCII-associated antigen presentation, resulting in potent cellular and humoral immune responses.

Importantly, the three LS exported proteins used in this study were selected based on a previous study, which showed their potential induction of CD8 T cells and their role in conferring sterile protection against infectious sporozoites when tested individually [5]. Despite the fact that the correlates of protection for each of the LS exported proteins in this viral vaccine were not tested, it will be the goal of future studies to identify the correlates of protection for the antigens presented by this vaccine. While not tested here, we hypothesize that anti-EXP-1, -TMP21, and/or -UIS3 CD8+ T-cells in the liver are responsible for the efficacy of our vaccine.

Herein, we have shown that the HSV-1(VC2) vaccine vector has significant potential to be used for anti-infectious disease vaccines and, specifically, immunization to protect against malaria infections. Our data support that using LS proteins as components of malaria vaccines is sufficient to establish sterilizing immunity in mouse models of malaria. Future work will be needed to test this malaria vaccine approach in other animal models of malaria infection, such as a non-human primate model. Successful testing in non-human primates could lead to human clinical trials using vaccines targeting LS antigens that could achieve the World Health Organization’s goal of malaria eradication by 2030.

## Figures and Tables

**Figure 1 vaccines-10-00300-f001:**
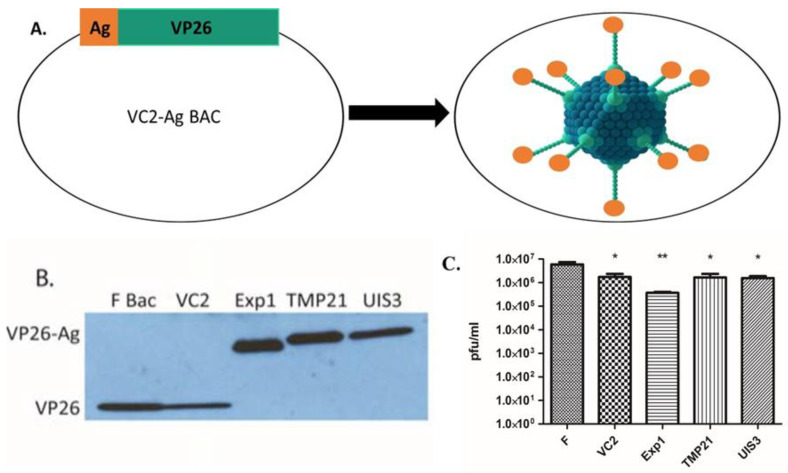
Construction of VC2-derived malaria vaccines. (**A**) Schematic representation of the fusion of plasmodium liver stage antigens with the VP26 viral capsid protein of VC2 strain. Three VC2-derived malaria vaccines (VC2-EXP1, VC2-TMP21 and VC2-UIS3) were independently generated using a red-mediated recombination of the VC2 bacterial artificial chromosome (BAC). (**B**) Fusion proteins easily detected in viral stocks. The protein expression of fused liver-stage antigens was assessed by Western blot analysis. The protein lysates from viral stocks (1 × 10^6^ pfu) were separated on SDS-PAGE, transferred to PVDF membrane, and probed with antibodies to HSV-1 VP26. Western blotting using anti-VP26 antibody demonstrated the expected increase in VP26 apparent molecular mass, consistent with the predicted fusion to respective malaria LS proteins (VP26 12kDa, Exp1VP26 26kDa, TMP21VP26 32kDa, and UIS3VP26 29kDa). (**C**) Fusion of malaria antigens does not significantly affect viral growth. Growth analysis was performed to determine whether fusion of LS antigens to VP26 affects the growth of these viruses. Vero cells were infected at an MOI of 0.01 and after 48 hours cells were harvested to assay viral titers. After plaque forming assay, we observed that the fusion of LS proteins with VP26 did not affect viral growth. Only VC2-EXP1 exhibited non-significantly lower growth in comparison to parental VC2. * *p* > 0.05, ** *p* > 0.005.

**Table 1 vaccines-10-00300-t001:** Intramuscular and subcutaneous vaccination with VC2-derived malaria vaccines induces sterile protection against intravenous infectious sporozoite challenge.

Groups (N)	Immunization Route (Intervals in Days)	Protected/Challenged ^1^
Group A: Pooled Vaccine (9)	SC (0, 21, 42)	9/9
Group B: Pooled Vaccine (8)	IM (0, 21, 42)	8/8
Group C: VC2 (5)	SC (0, 21, 42)	0/5
Group D: VC2 (5)	IM (0, 21, 42)	0/5
Group E: Naïve Control (5)	N/A	0/5

^1^ Virulent sporozoite challenge of the malaria vaccine and control groups was performed 12 weeks after the last vaccination dose by intravenous injection of 500 *P. yoelii* 17X-NL salivary gland sporozoites.

## Data Availability

Not applicable.

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
