# Peer review of "An Attenuated HSV-1-Derived Malaria Vaccine Expressing Liver-Stage Exported Proteins Induces Sterilizing Protection against Infectious Sporozoite Challenge"

_vaccines, 2022, doi:10.3390/vaccines10020300_

Round 1

Reviewer 1 Report

The paper entitled « An attenuated HSV-1-derived malaria vaccine expressing liver stage exported proteins induces sterilizing protection against virulent sporozoite challenge” by Rider at al. describes an interesting study for finding good vaccine candidates against malaria. The disease is a global health threat, especially in the developing world. The current artesunate monotherapy or artesunate-mefloquine combination therapy was associated with increase of resistance in Thailand and Cambodia. There is no highly effective vaccine, and the recent phase 3 clinical trials of the candidate RTS,S are disappointing. The authors in this study used the HSV-1 derived viral vector VC2 to express three different malaria antigens from the pre-erythrocytic stage and found complete protection through IM and SC immunizations at 15 days post-challenge. Overall, this paper is well written. Although the use of this attenuated vector as a vaccine strategy is not novel, the potential of these vectors to express liver-stage malaria parasite proteins as fusion with the capsid protein is unique. It will be interesting to have the contribution of each candidate vaccine and whether the pooled candidates contribute to any improvement. This is the beginning for these investigators to pursue this strategy and investigate in detail the associated immune and humoral responses. This has the potential to open a new field of study towards a new vaccine candidate against malaria. Some relatively major points need to be addressed to help in the rigor of the presentation of this manuscript.

General points:

-Figure 1A is irrelevant, and the authors might delete it.

-Please add the marker size (kDa) in Figure 2

-If the authors have antibodies against one of these proteins, please share the immunoblotting data

-Please add in materials and methods the amount of proteins loaded per well

-The introduction needs some restructuration. Please add a section on the current vaccine approaches under investigation against malaria. The first and second paragraph of the discussion should be moved to the introduction section

Minor points:

- There is no information about EXP1, TMP21, and UIS3. What are their roles in the pre-erythrocytic stages, and why were they chosen? Please add references. It will also help to add respective sequences fused to VP26 as supplement data

- Statistical analysis (p-values ) is needed in figure C even there is no significant growth between groups.

- Missing dot after reference [7].

Decision: Minor revisions

Author Response

Reviewer 3:

The paper entitled « An attenuated HSV-1-derived malaria vaccine expressing liver stage exported proteins induces sterilizing protection against virulent sporozoite challenge” by Rider at al. describes an interesting study for finding good vaccine candidates against malaria. The disease is a global health threat, especially in the developing world. The current artesunate monotherapy or artesunate-mefloquine combination therapy was associated with increase of resistance in Thailand and Cambodia. There is no highly effective vaccine, and the recent phase 3 clinical trials of the candidate RTS,S are disappointing. The authors in this study used the HSV-1 derived viral vector VC2 to express three different malaria antigens from the pre-erythrocytic stage and found complete protection through IM and SC immunizations at 15 days post-challenge. Overall, this paper is well written. Although the use of this attenuated vector as a vaccine strategy is not novel, the potential of these vectors to express liver-stage malaria parasite proteins as fusion with the capsid protein is unique. It will be interesting to have the contribution of each candidate vaccine and whether the pooled candidates contribute to any improvement. This is the beginning for these investigators to pursue this strategy and investigate in detail the associated immune and humoral responses. This has the potential to open a new field of study towards a new vaccine candidate against malaria. Some relatively major points need to be addressed to help in the rigor of the presentation of this manuscript.

General points:

-Figure 1A is irrelevant, and the authors might delete it.

Thank you for the suggestion.  We however feel that this figure really drives home one of the most innovative aspects of our approach:  fusion to of antigen to the minor capsid protein.

-Please add the marker size (kDa) in Figure 2 

Done.  We thank the reviewer for the suggestion.

-If the authors have antibodies against one of these proteins, please share the immunoblotting data

This is a wonderful suggestion.  Unfortunately, there are no antibodies that exist. We have recently created recombinant proteins that will be used to generate antibodies for future experiment.

-Please add in materials and methods the amount of proteins loaded per well.

Thank you for bringing this to our attention.  We did not determine protein concentration for this experiment as we were not trying to be quantitative but rather simply trying to detect the fusion protein.  We do however add the same amount of lysate (20ul) and this has been added to the materials and methods.  We note that as the infections are standardized the amount of viral proteins is expected to be fairly standardized across samples.

-The introduction needs some restructuration. Please add a section on the current vaccine approaches under investigation against malaria. The first and second paragraph of the discussion should be moved to the introduction section

Minor points:

- There is no information about EXP1, TMP21, and UIS3. What are their roles in the pre-erythrocytic stages, and why were they chosen? Please add references.

- It will also help to add respective sequences fused to VP26 as supplement data

Thank you. Protein sequences of fusion proteins have been added as supplementary information. 

- Statistical analysis (p-values ) is needed in figure C even there is no significant growth between groups.

Thank you. This has been added. 

- Missing dot after reference [7].

Thank you. This has been changed. 

Decision: Minor revisions

Reviewer 2 Report

Known in the field based on previous literatures:

  1. Malaria is one of the deadliest infectious diseases worldwide. Emergence of multi drug resistance signifies the need for novel targets/drugs to combat malaria parasites.
  2. Various research groups working on malaria vaccine are mainly targeting pre-erythrocytic, erythrocytic and transmission blocking stages, but none of them has achieved an effective vaccine. Among several potential vaccines under development that target the pre-erythrocytic stage of the parasites, RTS, S has shown the most promising results so far, but it also showed modest to low efficacy, indicating an urgent need for a safe, effective malaria vaccine.

In this manuscript authors reported following findings:

Authors performed very interesting work. They used the novel HSV-1-derived viral vaccine vector VC2, to express three different malaria parasite liver stage (LS) antigens.

  • Authors successful created three different recombinant malaria antigens. They used the novel HSV-1-derived viral vaccine vector VC2, to express three different malaria parasite liver stage (LS) antigens- EXP1, TMP21 and UIS3.
  • They pooled together these antigens/vaccines, and vaccinated mice prior to intravenous challenge with sporozoites. Authors reported that immunized mice generate sterile protection against virulent Plasmodium sporozoite in a mice model.

 Minor Concerns:

Really interesting work performed by the authors. Although authors claim 100% sterile protection in immunized mice but there are many unanswered questions which could be part of future study. The following minor suggestions if incorporated could help in the better understanding of the significance of the work.

Some of the minor comments are-

  1. What was the rationale to used three different antigens together without studying the effect of individual or combined vaccines?
  2. The genetic background of both host and parasites can greatly influence malaria disease severity, and almost identical genome led to different disease phenotypes. For examples, BALB/c mice infected with the parasites Plasmodium yoelii yoelii 17XL (YM) die within 7 days of post infection, whereas isogenic strain P. y. yoelii 17XNL recover from infection. Have you checked the vaccines effect using other strain like 17XL?
  3. Authors observed 100% sterile protection in immunized mice however control mice detected parasites by day 4 post sporozoite challenge. What happened to naïve control mice? Did you follow the parasitemia in control mice if yes what was the parasitemia at the time of termination??
  4. What was the reason to use only 500 P. yoelii sporozoites? Have you checked higher number or other number too?
  5. How did you decide the three immunizations at three-week interval?
  6. Please mention the molecular weight of these three proteins over western blot figure.
  7. Please check/correct the spelling of Preshant.

Author Response

  1. Malaria is one of the deadliest infectious diseases worldwide. Emergence of multi drug resistance signifies the need for novel targets/drugs to combat malaria parasites.
  2. Various research groups working on malaria vaccine are mainly targeting pre-erythrocytic, erythrocytic and transmission blocking stages, but none of them has achieved an effective vaccine. Among several potential vaccines under development that target the pre-erythrocytic stage of the parasites, RTS, S has shown the most promising results so far, but it also showed modest to low efficacy, indicating an urgent need for a safe, effective malaria vaccine.

In this manuscript authors reported following findings:

Authors performed very interesting work. They used the novel HSV-1-derived viral vaccine vector VC2, to express three different malaria parasite liver stage (LS) antigens.

  • Authors successful created three different recombinant malaria antigens. They used the novel HSV-1-derived viral vaccine vector VC2, to express three different malaria parasite liver stage (LS) antigens- EXP1, TMP21 and UIS3.
  • They pooled together these antigens/vaccines, and vaccinated mice prior to intravenous challenge with sporozoites. Authors reported that immunized mice generate sterile protection against virulent Plasmodium sporozoite in a mice model.

 Minor Concerns:

Really interesting work performed by the authors. Although authors claim 100% sterile protection in immunized mice but there are many unanswered questions which could be part of future study. The following minor suggestions if incorporated could help in the better understanding of the significance of the work.

Some of the minor comments are-

  1. What was the rationale to used three different antigens together without studying the effect of individual or combined vaccines?

Response: In a previous study from Aly lab, in a collaboration with the Weiner lab (Wistar Institute), these three antigens were studied among 5 different exported liver stage antigens as DNA vaccine candidates (Reeder et al, 2020_Vaccines). Immunizations with different combinations with all of the antigens confirmed that these three antigens elicited significant cellular immune responses and also conferred sterile protection against sporozoite challenge.

  1. The genetic background of both host and parasites can greatly influence malaria disease severity, and almost identical genome led to different disease phenotypes. For examples, BALB/c mice infected with the parasites Plasmodium yoelii yoelii 17XL (YM) die within 7 days of post infection, whereas isogenic strain P. y. yoelii 17XNL recover from infection. Have you checked the vaccines effect using other strain like 17XL?

Response: Sporozoite invasion and liver stage development pattern are considered to be identical between 17X-L and 17X-NL strains. Since the immunity generated with these immunizations is targeting pre-erythrocytic stages and not blood stages, we do not anticipate that the sporozoite challenge with 17X-L will lead to different results than with 17X-NL sporozoites.

  1. Authors observed 100% sterile protection in immunized mice however control mice detected parasites by day 4 post sporozoite challenge. What happened to naïve control mice? Did you follow the parasitemia in control mice if yes what was the parasitemia at the time of termination??

Response: Since the challenge was done with 17X-NL sporozoites, we didn’t expect that blood stage infection will lead to death or pathology of any of the naïve controls or the VC2 control mice.

Nevertheless, we followed the infection of the naïve and VC2 controls, and at the time of termination (day 15 post sporozoite challenge) they were still highly positive, between 8 and 20% but their parasitemias % were already in decline.

  1. What was the reason to use only 500 P. yoelii sporozoites? Have you checked higher number or other number too?

Response: In our previous studies, we established that 20 salivary gland sporozoites of P. yoelii is able to infect mice reliably (Aly et al, 2008). In our latest study from 2020, (Reeder et al, 2020), we used 250 salivary gland sporozoites of 17X-NL strain as a challenge dose. We believe that 250-500 is a more realistic dose, with relation to the number of sporozoites inoculated during an infectious bite, of challenge with salivary gland sporozoites that will lead to 100% infection in naïve control mice.

  1. How did you decide the three immunizations at three-week interval?

Response: In our previous study from 2020, we successfully used three-week intervals to generate protective cellular immunity against a DNA vaccine. Since this is an attenuated viral vaccine, we anticipated that a three-week interval will be suitable for the generation of adaptive cellular immune responses.

  1. Please mention the molecular weight of these three proteins over western blot figure.

Response: We added the molecular weights of the fusion proteins to the figure legend, which were 26kDa, 32kDa and 29 kDa for EXP1, TMP21, and UIS3 respectively.

  1. Please check/correct the spelling of Preshant.

Response: We corrected this spelling mistake to Prashant.

Reviewer 3 Report

I have added suggestions to the PDF and these are attached.

My overarching criticism is that by combining all three antigens in one experiment, there is no way of determining which of the antigens actually provide protection. It is of my opinion that this work cannot be published until experiments are carried out that test the efficacy of each of the antigens. In addition, experiments need to take place to determine the mode of protection - antibodies, T cells or a combination of both. Ultimately, does de-escalation would also help understand the power of each antigen to promote an effective immune response.

Author Response

My overarching criticism is that by combining all three antigens in one experiment, there is no way of determining which of the antigens actually provide protection. It is of my opinion that this work cannot be published until experiments are carried out that test the efficacy of each of the antigens.

Response: In a previous study from Aly lab, in a collaboration with the Weiner lab (Wistar Institute), these three antigens were studied among 5 different exported liver stage antigens as DNA vaccine candidates (Reeder et al, 2020_Vaccines). Immunizations with different combinations with all of the antigens confirmed that these three antigens elicited significant cellular immune responses and also conferred sterile protection against sporozoite challenge.

In addition, experiments need to take place to determine the mode of protection - antibodies, T cells or a combination of both. Ultimately, does de-escalation would also help understand the power of each antigen to promote an effective immune response.

Response: These are excellent suggestions. However, these detailed experiments are out of the scope for this brief report and we are certainly planning these detailed experiments for a follow up study.

We feel that our approach to generating a malaria vaccine is highly innovative and will be of great interest and impact as it is currently written. As evidence that the suggested experiments are not necessary to report a novel malaria vaccine we submit our previous publication in Vaccines. 2020 Jan 10;8(1):21. doi: 10.3390/vaccines8010021.

Round 2

Reviewer 3 Report

As I mentioned in my first review, the protection provided by each individual antigen needs to be assessed.

Author Response

We understand the concern of the expert reviewer, and we agree that the rationale of selecting these three antigens in a combination vaccination was not clear enough.

We have now explained in the introduction and discussion that each of these selected antigens were shown to induce cellular immune responses and partial sterile immunity when they were tested separately (Reeder et al, 2020).

However, these antigens showed complete sterile protection when they were tested in a combined vaccination of all these antigens (Redder et al, 2020).

We have highlighted these changes in the revised version of the manuscript.